# The Use of Micro-Ribbons and Micro-Fibres in the Formulation of 3D Printed Fast Dissolving Oral Films

**DOI:** 10.3390/ph16010079

**Published:** 2023-01-05

**Authors:** Marwan Algellay, Matthew Roberts, Lucy Bosworth, Satyajit D. Sarker, Amos A. Fatokun, Touraj Ehtezazi

**Affiliations:** 1Centre for Natural Products Discovery, School of Pharmacy and Biomolecular Sciences, Liverpool John Moores University, Byrom Street, Liverpool L3 3AF, UK; 2Department of Eye and Vision Science, Institute of Life Course and Medical Sciences, University of Liverpool, 6 West Derby Street, Liverpool L7 8TX, UK

**Keywords:** three-dimensional printing, fast dissolving oral films, fused deposition modelling, chitosan, cellulose

## Abstract

Three-dimensional printing (3DP) allows production of novel fast dissolving oral films (FDFs). However, mechanical properties of the films may not be desirable when certain excipients are used. This work investigated whether adding chitosan micro-ribbons or cellulose microfibres will achieve desired FDFs by fused deposition modelling 3DP. Filaments containing polyvinyl alcohol (PVA) and paracetamol as model drug were manufactured at 170 °C. At 130 °C, filaments containing polyvinylpyrrolidone (PVP) and paracetamol were also created. FDFs were printed with plain or mesh patterns at temperatures of 200 °C (PVA) or 180 °C (PVP). Both chitosan micro-ribbons and cellulose micro-fibres improved filament mechanical properties at 1% *w*/*w* concentration in terms of flexibility and stiffness. The filaments were not suitable for printing at higher concentrations of chitosan micro-ribbons and cellulose micro-fibres. Furthermore, mesh FDFs containing only 1% chitosan micro-ribbons disintegrated in distilled water within 40.33 ± 4.64 s, while mesh FDFs containing only 7% croscarmellose disintegrated in 55.33 ± 2.86 s, and croscarmellose containing films showed signs of excipient scorching for PVA polymer. Cellulose micro-fibres delayed disintegration of PVA mesh films to 108.66 ± 3.68 s at 1% *w*/*w*. In conclusion, only chitosan micro-ribbons created a network of hydrophilic channels within the films, which allowed faster disintegration time at considerably lower concentrations.

## 1. Introduction

Fast-dissolving oral films (FDFs) are a new way to boost consumer acceptance since they dissolve quickly in the mouth, and usually are administered without water. FDFs are suitable for paediatric and geriatric patient populations, where the difficulty of swallowing larger oral dosage forms is eliminated. FDFs can also be useful for patients who have swallowing difficulties (dysphagia). It is estimated that approximately 1 in 25 adults has swallowing problems [1]. Three-dimensional printing (3DP) technology, also known as additive manufacturing (AM), is based on computer-aided design (CAD) and is capable of selectively placing printing materials layer-by-layer to generate pieces to assemble a final product with predesigned geometry [2]. 3DP has gained a lot of attention as a unique pharmaceutical manufacturing process because of its particular benefits over traditional production approaches, such as the ability to achieve desirable and variable medicinal doses [3]. The main advantage of 3DP for pharmaceutical oral films is producing multi-layered or mesh FDFs, [4] and achieving personalised FDFs [5]. 3DP allows to adjust disintegration time of FDFs by printing in a mesh-design [4]. Additionally, drug crystallisation may be prevented in oral films by 3DP compared to solvent casting method [6]. FDFs were prepared by a gel 3DP with disintegration time of 3 s when sodium carboxymethylcellulose was included in the formulation at 5% *w*/*w* concentration [7]. In another approach, Elbadawi et al., 2021 employed dual-nozzle pressure-assisted microsyringe 3DP for fabrication of FDFs [8].

The fused deposition modelling (FDM) 3DP appears to be a promising technique to produce oral drug forms for precision medicine [9]. FDM works by melting the applied materials, such as medications and other essential excipients in the printer head, into a semisolid state, which is then extruded into a fine filament on a platform (print bed). The molten material is deposited side-by-side by the parallel (rastered) back and forth movement of the printer head to create a 2D pattern. Then, the z-axis movement of the platform or printer head deposits the molten filament layer-by-layer creating a 3D object [10,11]. When it comes to cost-effectiveness and production speed, FDM is one of the most promising rapid prototyping methods. About half of all 3D printers falls under this group [12]. However, due to the generally low mechanical qualities of 3D-printed components, the application of fused filament fabrication or FDM printing may be limited [13]. Typically FDM requires the material to be fed into printer head in the form of a filament [11], or powder [14]. In order to ensure the filament’s suitability for FDM 3DP, it is necessary to determine the optimal mechanical properties of the filament, such as elasticity, stiffness, and brittleness [6]. One solutions is the use of long molecular weight polymers [4], while this may increase the disintegration time, which is not a desired property for FDFs [4].

The exceptional features of nano-biomaterials make them ideal for a wide range of biomedical applications, including improved tissue/organ regeneration [15]. Because of their high aspect ratio and a very high surface-to-volume ratio, nanofillers outperform macrofillers. Furthermore, evenly distributing these nanofillers into the host matrix allows them to provide a wide interfacial area per volume, which improves the mechanical characteristics of the polymers [16]. 3D printed specimens revealed an improvement in stiffness when crystalline nanocellulose was incoporated into polyvinyl alcohol (PVA) in the range of 2–10% *w*/*w* [17]. The mechanical properties also improved for FDM printed parts, when carbon nanotubes (CNT) [18], or maghemite (γ-Fe_2_O_3_) nanoparticles [19] were incorporated the modulus of elasticity was enhanced to 30% with 5% *w*/*w* addition of CNT into PLA matrix [20].

Chitosan, a natural polysaccharide [21], can be formed into a micro/nano-fibre structure with a diameter ranging from 50 to 500 nm [22]. As an example of another natural product, cellulose-based composites are considered ecologically friendly [23]. Nanofibrillated cellulose composites improve tensile properties of epoxy resin films [24], provide excellent mechanical qualities, reinforcing capabilities, low density, thermal stability and biodegradability [25]. They can also be employed in polymer bio-nanocomposites as nanofillers or matrices [26]. 

The present study investigated whether the use of different types of pharmaceutical nanofibre polymers could allow the use of lower molecular weight polymers for preparing FDFs by the FDM 3DP. The research plan is demonstrated in Figure 1. This was to achieve 3DP FDFs with desired properties such as matching disintegration time with FDFs prepared by the solvent casting method, lowering printing temperatures, and obtaining acceptable mechanical properties. The main objectives of this study were to: (1) couple FDM 3D printing with the hot melt extrusion (HME) technology to print FDFs by using filaments loaded with nanofibres/microfibres of chitosan or cellulose, (2) screen different grades of nanofibres/microfibres suitable for 3D printing and evaluate their effects on the physical and mechanical properties of FDFs, and (3) study the drug release profiles, cytotoxicity, and disintegration time of 3DP films.

## 2. Results

### 2.1. Morphology of Chitosan Micro-Ribbons and Cellulose Microfibres

Figure 2 presents the SEM images of freeze-dried chitosan and Figure 3 shows SEM images of micro-fibrillated cellulose with different grades (C500, C1000, C2000). SEM data indicated that freeze dried chitosan formed ribbons with the width typically around 20 µM. Therefore, freeze-dried chitosan was designated as chitosan micro-ribbons (Chi-MRs) in this paper. Figure 3 illustrates that micro-fibrillated cellulose formed a mixture of mainly micro-fibres and some micro-ribbons. Therefore, micro-fibrillated cellulose was denoted as cellulose-microfibres (Cel-M).

### 2.2. Formation of Filaments and Films

Higher concentrations of Chi-MRs made the surface of filaments unsmooth, making it difficult to feed to the printer head. As a result, films were not printed. Appendix A shows the extensive rough surface of the filament from formulation FS3, compared to the smooth filament surface from formulation F2. On the other hand, PVA filaments were able to be printed in formulation containing Cel-M up to 3% *w*/*w*. Croscarmellose was removed in formulation F3, and filaments with white colours were obtained, suggesting scorching of croscarmellose during extrusion. PVA was replaced with PVP 40K in formulations F18 to F23. Inclusion of PVP 40K reduced the extrusion temperature from 170 °C to 90 °C. PVP40K formed filaments on its own (formulation FS6), however, these filaments were unable to load into the either Prusa printer or the RS pro. These filaments were squeezed and crushed (Appendix A) between the printer’s head rollers (Appendix A). Preliminary studies showed that the inclusion of only Cel-M 1000 grade with the concentrations in the range of 5–10% *w*/*w* did not lead to printable filaments containing PVP40K. Therefore, PEO 100K was employed to produce printable filaments (Formulations F18 to F23). PVP10K did not produced filament either on its own or with PEO 100K (Appendix A, FS15 to FS17). Filaments were not produced, even by adding 5% *w*/*w* C1000 (i.e., using longer microfibres at a relatively high concentration) to a formulation containing PVP10K (Appendix A, FS17).

All formulations in Table 1 formed films. Figure 4 presents photos of typical films. RS pro printer had soft feeding rollers inside the printer head, which allowed printing films of filaments containing PVP40K only for certain formulations (F18 to F23). The other films were printed using the Prusa 3D printer. None of the formulations in Appendix A were 3D printed (either the Prusa or RS pro 3D printers), apart from formulations FS1 and FS2. Comparing Table 1 and Appendix A shows that formulations F2, F3 and F4 which contained Chi-MRs in the range of 1–2% *w*/*w* were printable. Formulations F3 and F4 did not contain croscarmellose. Figure 5A presents SEM image of a printed film from formulation F1 and Figure 5B shows SEM image of formulation F2 containing Chi-MRs. The film without Chi-MRs (F1) showed smooth and regular surface, while the film containing Chi-MRs presented slight rough surface and irregular bands. These images suggest that Chi-MRs caused inconsistent flow of molten polymer from the printer head. 

### 2.3. Weight, Thickness Disintegration Time and Content Uniformity of FDFs

Figure 6A–C present disintegration time, dimensions and weights of the films, respectively. Figure 6A shows disintegration time for both plain and mesh films. Formulation FS2 (PVA and paracetamol only) had disintegration time of 247 ± 5 s and this was reduced to 125 ± 6 by printing the film in mesh shape. While formulation F1 demonstrated disintegration times of 120 ± 6 s and 55 ± 3 s with 7% *w*/*w* croscarmellose, for plain and mesh films, respectively. Adding 1% Chi-MRs to formulation F2 changed the disintegration time to 106 ± 5 s and 69 ± 8 s for plain and mesh films, respectively. Removing croscarmellose in formulation F3, reduced the disintegration time to 98 ± 5 s and 40 ± 5s for plain and mesh films, respectively (Figure 6A). Using higher amounts of Chi-MRs increased disintegration time of formulation F4 compared to formulation F3. On the other hand, the inclusion of Cel-M grade 500 at 1% *w*/*w* (formulation F5) increased the disintegration time to 223 ± 5 s for plain films compared to formulation F3 (98 ± 5). A similar trend was observed for mesh films of formulations F5 and F3. Formulation F17 was similar to the F3, but instead it contained 1% *w*/*w* chitosan as powder. The disintegration time was 239 ± 18 s and 118 ± 7 s for plain and mesh films, respectively. The disintegration time was considerably higher for F17 formulation compared to F3 formulation. This suggests that chitosan micro-ribbons form a hydrophilic network inside the film contributing to the disintegration of the film. Only formulation F19 achieved disintegration times of 52 ± 6 s and 42 ± 4 s for plain and mesh films, respectively. This formulation contained 30% *w*/*w* PEO 100K, 30% *w*/*w* PVP 40K, and 7% *w*/*w* croscarmellose. Surprisingly, formulation F20 was similar to the F19, but it contained 0.2% *w*/*w* Chi-MRs, and the disintegration times increased to 115 ± 7 s and 83 ± 9 s for plain and mesh films, respectively. 

Formulation F5 to F13 had increasing concentrations of Cel-M (all grades), and these showed much longer disintegration time for both plain and mesh films compared to F3 formulations films (Figure 6A). Surprisingly, the disintegration time even became longer when croscarmellose was added into the formulations containing Cel-M 500 (formulations F14, F15, F16). These observations suggest the formation of strong bonds between Cel-M and croscarmellose.

Figure 6B demonstrates the thickness of both plain and mesh films. The thickness of PVA films remined relatively unchanged for both plain and mesh films by adding micro-ribbons. A similar trend was also observed for PVP 40K films. However, the thickness of PVA films were less than PVP 40K films.

Figure 6C presents the weights of both plain and mesh films. The weights of the films were affected by adding micro-ribbons. The weight of films decreased from 90 ± 3 mg (formulation F1) to 78 ± 1 mg (formulation F2), by adding 1% Chi-MR. This suggests the slow movement of molten filament from the printer head to the printing platform. This may be explained by the increase in the viscosity of the molten polymer due to the presence of Chi-MR. However, further addition of Chi-MR increased the weights of the films (Formulation F4 compared to formulation F2). In addition, increasing Cel-M grade decreased the weight of the films (formulations F21, F22, F23).

Figure 7 presents the percent of nominal dose for filaments, mesh films, and plain films of formulations F1, F2, F3, F8, F19, F20, F21, F22, and FS2. Appendix A shows the weights, amounts of measured paracetamol, and percent of nominal dose for the formulations. It can be seen from Figure 6 that the amounts of paracetamol varied in the range of 85–115% of the nominal dose for formulations F8, F19, F20, F21 and F2. While the amounts of paracetamol dropped below 90% for both plain and mesh films of F3 formulation and only for mesh films of F1 and F2 formulations. Filaments showed similar or higher amounts of paracetamol compared to the films, perhaps due to lower processing temperature.

### 2.4. DSC and FTIR

Figure 8A presents the DSC thermograms for powder mixtures of formulations F1, F2, F3, F5, F8 and F11. The thermograms of pure paracetamol, PVA and croscarmellose are also included in Figure 8A. The peak of the DSC curve for pure paracetamol was found at 178 °C. Physical mixtures of formulation showed only one peak around paracetamol melting temperature. There were 2–3 °C peak changes, probably due to interactions between paracetamol and the polymers. The DSC thermograms did not show any peaks (including paracetamol) for the filaments and films of the formulations (Figure 7B,C, respectively). This implies that throughout the hot-melt extrusion and printing processes, paracetamol diffused in the molten polymer matrix to generate a homogenous solid dispersion, and/or there were changes in paracetamol molecules. On the other hand, powder mixtures of formulations F18, F19, F21 showed only one endothermic peak at 60 °C, corresponding to melting point of PVP 40K or PEO 100K. 

Figure 9A and Figure 8B present the FTIR spectra of formulations F2 and F3 for powder mixture, filament, and film, respectively. The spectra suggest chemical changes, in particular between 3300–2920 cm^−1^ by formation of filament and film compared to the powder mixture. These suggest possible chemical changes in paracetamol molecules. Figure 8C,D present the FTIR spectra of formulations of F19 and F20 for powder mixture, filament, and film, respectively. The FTIR spectra do not change considerably for formulation F19, but the FTIR spectra of formulation F20 change between 3000 cm^−1^ and 3500 cm^−1^. In addition, a peak appears at 1800 cm^−1^ for formulation F19 for the film (shown by a circle in Figure 9C). This suggests the formation of an ester bond in the formulation. Figure 9E demonstrates the FTIR spectra of formulation F21 for powder mixture, filament, and film. There are changes between 3000 cm^−1^ and 3500 cm^−1^ by making filament and film, as well as appearance of a peak at 1800 cm^−1^. These observations suggest preventing certain reactions by chitosan molecules in the formulation during printing of the films. Appendix A presents the FTIR spectra of paracetamol, PVA and croscarmellose for comparisons.

### 2.5. In Vitro Release Profiles

Figure 10A presents in vitro drug release profiles of formulation F1, F3, F4, F5, F8, F9, F10, F19, and F20 for plain films and Figure 10B presents the release profiles of the same formulations but for mesh films. Plain F20 film showed the fasted drug release rate with minimal error bars. This formulation contained only 0.2% Chi-MRs. Formulation F19 showed slower drug release rate compared to formulation F20, and F19 did not contain Chi-MR. Figure 10A also shows that the plain films of formulation F5 had slowest drug release rate, which contained 1% *w*/*w* Cel-M C1000, compared to the other plain films including formulation F8, which also contained 1% *w*/*w* Cel-M 1000, while the plain-films of F5 had much faster disintegration time compared to plain-films of F8 (Figure 6A). The other formulations showed similar drug release profiles. Plain F2 formulation showed the largest error bars (up to 18%), which contained 1% *w*/*w* Chi-MR.

As expected, mesh films showed faster drug release compared to plain films. All formulations showed smaller error bars compared to plain films. Mesh formulation F2 showed the fastest drug release rate, while mesh F1 formulation showed the slowest drug release rate, suggesting that hydrophilic network of Chi-MRs facilitated exposure of active ingredients to the release media. Surprisingly, removing croscarmellose in formulation F3 reduced drug release rate.

### 2.6. Mechanical Properties of Filaments and Films 

The stiffness results of the filaments are shown in Figure 11A, which compares printable filaments with non-printable ones. Printable filaments had stiffness greater than 100 kg/mm^2^%. Non-printable filaments presented stiffness as much as printable filaments. For example, formulation F3 contained 1% *w*/*w* Chi-MRs, while increasing the Chi-MR level to 2% *w*/*w* in formulation FS3 with similar stiffness but rendered the filament to non-printable one. Further increasing Chi-MR level to 3% in formulation FS4 (non-printable filament) reduced the stiffness of the filament, but was still much higher than F22, which was a printable filament. These observations indicate that part of non-printable filaments failed to print either possibly due to extensively increased viscosity of the molten polymer preventing extrusion from the printhead nozzle, or possibly the nozzle became blocked due to the fibres in the filament. Interestingly, removing croscarmellose decreased the filament’s stiffness for formulation F3 compared to formulation F2.

The results of three-point bend (flexibility) and resistance of filaments are given in Figure 11B,C, respectively. Printable filaments had flexibility greater than 6.5 kg/mm^2^%. All filaments containing PVP showed less flexibility compared with PVA filaments. This could explain the reason why PVP filaments were not loaded to Prusa printer as filaments squashed with lower flexibility and could not resist the gear mechanical stress during loading process (Appendix A). Adding 1% Chi-MRs to formulation F2 elevated flexibility (three-point bend test) from 8.13 ± 1.61 to 13.09 ± 3.40 kg/mm² %. The flexibility of formulation F5 (containing 1% *w*/*w* C500 Cel-M) reached maximum at 18.36 ± 2.06 kg/mm² %. However, further adding C500 Cel-M in formulation F6 and F7 reduced the flexibility of the films. Furthermore, flexibility of formulation FS3 increased from 14.42 ± 7.46 to 24.51 ± 7.40 kg/mm²% by increasing Chi-MR from 2% *w*/*w* to 3% *w*/*w* (formulation FS4). Similar trends were observed for the resistance test (Figure 11C). This was much pronounced for formulations FS3 and FS4.

Figure 12A presents the strength of printed films. The addition of either Chi-MRs or Cel-Ms increased the strength of films. Figure 12B presents the elongation of the films, and again the addition of micro-ribbons increased the mean elongation at break. Moreover, cellulose micro-fibres provided higher elongation compared to Chi-MRs.

### 2.7. Cell Toxicity Studies

Figure 13 presents the percentage cell viabilities following 48 h treatment of the HeLa cells with the oral films and micro-ribbons. All formulations and ingredients showed cell viability values greater than 90% suggesting the films and micro-ribbons were non-cytotoxic to HeLa Cells.

## 3. Discussion

This study found that the addition of micro-ribbons or micro-fibres changed the mechanical properties of filaments (either increasing or decreasing the stiffness). The addition of only Chi-MRs at low concentration considerably decreased considerably the disintegration of only PVA containing films. Similar disintegration time was achieved by changing the PVA to PEO and PVP, with no-addition of micro-ribbons. This paper achieved its aim only for films containing PVA, i.e., producing FDFs with disintegration time less than one minute by using chitosan micro-ribbons. However, the inclusion of micro-ribbons increased the disintegration time for films containing PEO and PVP. Our results also indicated that non-printable PVA filaments containing micro-ribbons (FS3-FS5) did not lack desired mechanical properties. This might be due to changes in the viscosity of the molten polymer and/or blockage of the printer nozzle by micro-ribbons. Further studies are required to determine the rheological behaviour of molten polymers with micro-ribbons, as well as the use of printers with nozzles greater than 0.4 mm. FTIR spectra showed modification in the paracetamol fingerprint, indicating possible drug degradation, which was also reflected in the content uniformity test. In addition, FTIR showed the presence of an extra peak at 1800 cm^−1^ by using PVP, which was disappeared when chitosan was added into the formulation. This may suggest possible chemical reactions between paracetamol and other formulation ingredients, which might have been inhibited by chitosan. Our FTIR data showed a change in the spectrum between filaments and films. For example, the FTIR spectrum of formulation F7 filament suggested the formation of further ester groups. Although this was not observed in our previous publication [4], thermal degradation of 3DP thermoplastics has been reported [27].

The presence of chitosan as micro-ribbons was essential to reduce the disintegration time, while chitosan as a powder could not provide this function. Increasing the contents of micro-ribbons made printing difficult, perhaps not only by increasing the viscosity [28] of molten polymer in the printer head [29], but also, by creating a rough surface on the surface of the filament, making it difficult for the filament to pass through the gear mechanism in the printer head.

In this study, the formation of a filament was required with desirable mechanical properties for printing the films. As a result, a number of formulations were excluded. The formulation F19 achieved the fastest disintegration time of 52 ± 6 s. This formulation did not contain micro-ribbons. A previous work reported disintegration time of 17.7 ± 1.5 s for orodispersible films made by direct printing [30]. In addition, Janigová et al., 2022 developed orodispersible films by semisolid extrusion 3DP with subsequent drying of films, which achieved disintegration time as short as 2.6 ± 0.32 s [31]. Using holt melt extrusion 3DP produced orodispersible oral films with disintegration time of 34 ± 14 s [32]. Therefore, further studies are required by using direct printing to investigate those formulations that could not form desired filaments in our study.

In this work, PVP and PVA polymers were chosen as the FDFs matrix forming agent, because of their water solubility and capacity to make filaments by hot melt extrusion [6]. Okwuosa et al. also found that printing was not possible with PVP40K only filaments [33]. We added PEO100K to prepare stronger filaments for printing, but the inclusion of cellulose micro-ribbons could not produce suitable filaments for printing. In this study, we found that hardware played an important role in printing filaments. We also found printable filaments with flexibility greater than 6.5 kg/mm^2^ using the Prusa 3D printer, and stiffness greater than 100 kg/mm^2^, which is in agreement with the work of Xu et al., 2020, who determined minimum filament stiffness of 80 kg/mm^2^ for Prusa I3 MK3S 3D printer [34].

In our work, we chose two 3D forms for film printing. Jamróz et al. reported that PVA film disintegrated at 27.5 ± 4.23 s and aripiprazole loaded PVA film developed a longer disintegration time of 43.00 ± 1.00 s, due to the hydrophobic nature of the medicinal substance and the smaller pore size of films [6]. However, our PVA and paracetamol only formulation (FS2) had a disintegration time of 247 ± 5 s and this was reduced to 125 ± 6 by printing the film in mesh shape. Formulation F3 contained croscarmellose and the disintegration time further decreased to 120 ± 6 s for plain film and 55 ± 3 s for mesh film. The differences between our findings and observations made by Jamróz et al., 2017 might be explained due to different methods of preparing PVA filaments (PVA was mixed and moistened with ethanol) [6].

Many factors affect the release of an active ingredient from polymeric films, like the solid state of the drug, the wettability by a hydrophilic carrier and excipients [35]. Satyanarayana and Keshavarao found that FDFs released 90% of loaded drug less than 240 s, when films were prepared by solvent casting method and hydroxypropyl methylcellulose was employed as the main polymer [36]. While our FDFs released 90% of drug within 30 min. The difference might be explained partly due to fast disintegration time of the FDFs prepared by solvent casting method (14 ± 1 s) compared to our films.

Chitosan nanoparticles did not show cytotoxicity at 200 µg/mL concentration against HeLa cells [37]. Our formulations containing chitosan micro-ribbons also did not present cytotoxicity towards HeLa cells even at concentrations as high as 1000 mg/mL. Furthermore, PVP did not show cytotoxicity against BV2, NIH-3T3, and SH-SY5Y cells [38]. Similarly, FDFs containing PVP did not show cytotoxicity against cells.

## 4. Materials and Methods

### 4.1. Materials

Paracetamol, polyvinyl alcohol (PVA, Mw 89,000–98,000 D, 99% hydrolyzed), chitosan (low molecular weight), polyvinylpyrrolidone (PVP, Mw 40,000 and 10,000 D) polyethelene-oxide (PEO, Mw 100,000 and 200,000 D) and sodium triphosphate pentabasic (TPP) were purchased from Sigma-Aldrich (Dorset, UK). Croscarmellose sodium was acquired from Merck-chemicals (Darmstadt, Germany). Cellulose microfibres (Celova^®^) with three different surface areas (C500, C1000 and C2000 mm^2^/g) were gifted from Healthy Suppliers Weidmann fibre technology (Rapperswil, Switzerland). Cell culture reagents (Dulbecco’s Modified Eagle (DMEM), foetal calf serum, L-Glutamine, antibiotic-antimycotic solution (penicillin/streptomycin/amphotericin B), and recombinant trypsin solution (TrypLE), were obtained from Thermo Fisher Scientific. The MTT dye (3-(4,5-dimethylthiazol-2-yl)-2 (UK) was purchased from Sigma-Aldrich (Dorset, UK).

### 4.2. Preparation of Chitosan Micro-Ribbons

Chitosan micro-ribbons (Chi-MRs) were prepared by the ionic gelation of chitosan by TPP as described previously [39]. Chi-MRs were prepared by adding dropwise, 3.6 mL of TPP solution (840 mg TPP per 100 mL distilled water) into 9 mL of chitosan solution (2.4 g of chitosan per 100 mL o f 1% *w*/*v* acetic acid), while stirring using a magnetic stirrer. The suspension were centrifuged in centrifuge tubes for 90 min at 3500 rpm. The sediment was then freeze dried at −55 °C and 0.16 mbar vacuum pressure for 7 days.

### 4.3. Preparation of Filaments

Table 1 presents formulation compositions. Further formulation compositions also investigated are presented in Appendix A. Powders of excipients and active ingredient (paracetamol) were introduced into the Tubula-mixer (Type 2B; WAB, Muttenz, Switzerland) and blended at the speed of 42 rpm for 15 min to obtain a homogenised mixture [4]. The powder mixture was fed to the single-screw hot melt extruder (Noztek pro^®^, Appendix A) that was rotating at 30 rpm with custom-made rod-shaped aluminium die (ø = 1.70 mm), targeting a filament with final diameter of 1.75 ± 0.05 mm. PVA filaments were prepared in the temperature range of 170–180 °C, and PVP filaments were produced in the temperature range of 90–140 °C. The filament diameter was determined every 5 cm. Each formulation was evaluated three times. The extruder was cleaned from powder residues after each formulation replica, and also the barrel was cleaned using a brush. The screw of the extruder was cleaned initially under running water, and then was subjected to sonication in water for 30 min. This was to avoid any cross-contamination. Preliminary work showed uniform distribution of Chi-MRs.

**Table 1 pharmaceuticals-16-00079-t001:** The weight percentage compositions of various ingredients in formulations of FDFs. 3D printing of FDFs was achieved with the filaments of these formulations.

Formulation	PCM	PVP 40K	PEO 100K	PVA	CCS	Chi-MR	C500	C1000	C2000	Chi
**F1**	30	-	-	63	7	-	-	-	-	-
**F2**	30	-	-	62	7	1	-	-	-	-
**F3**	30	-	-	69	-	1	-	-	-	-
**F4**	30	-	-	68	-	2	-	-	-	-
**F5**	30	-	-	69	-	-	1	-	-	-
**F6**	30	-	-	68	-	-	2	-	-	-
**F7**	30	-	-	67			3			-
**F8**	30	-	-	69	-	-	-	1	-	-
**F9**	30	-	-	68	-	-	-	2	-	-
**F10**	30	-	-	67				3		-
**F11**	30	-	-	69	-	-	-	-	1	-
**F12**	30	-	-	68	-	-	-	-	2	-
**F13**	30			67					3	
**F14**	30	-	-	62	7			1		
**F15**	30	-	-	61	7			2		
**F16**	30	-	-	60	7			3		
**F17**	30	-	-	69						1
**F18**	30	40	30	-	-	-	-	-	-	-
**F19**	30	33	30		7					
**F20**	30	33	29.8		7	0.2				
**F21**	30	33	25		7		5			
**F22**	30	33	25		7			5		
**F23**	30	33	25		7				5	

### 4.4. 3D Printing of FDFs

The films were manufactured mainly using the FDM Prusa^®^ i3 MK3S (Appendix A) 3D printer (Prague, Czech Republic). In addition, the RS PRO^®^ IdeaWerk (Appendix A) 3D Printer (RS Components Ltd., Northants, UK) was examined for certain formulations (containing PVP) due to different gearing mechanism in this 3D printer head. The films were designed using the SolidWorks^®^ 3D CAD software (Dassault Systèmes SolidWorks Corp., Waltham, MA) and saved as stereolithographic format (stl). Square plain films had dimensions of 20 mm in length and width, and 0.2 mm thickness. The stl files were exported to the PrusaSlicer software (version 2.3.3; Prague, Czech Republic) to use with Pursa^®^ i3 MK3S printer. The slicer software was also used to create a mesh film design with 50% triangle infill. Appendix A presents the designs of plain and mesh films by the Prusa slicer software. The printing parameters were 100% infill for plain films and 50% triangles infill for mesh films, two shells, 0.10 mm layer height, and extruder temperature of 200 °C. The non-extrusion travel move speed was 60 mm/s, with an infill travel speed of 30 mm/s and a printer bed temperature of 50 °C. Sticky masking blue tape (3M™) was utilised to help the adherence of printed film to the printer bed. Printing time was about 3 min for plain films, whereas the printing time was 2 min for mesh films. The printer head moved diagonally for printing the plain film using the Prusa printer, while the printer head moved in parallel with the sides of the film using the RS Pro printer. Both printers had nozzles with 0.4 mm diameters. To print with the RS Pro printer, the MakerWave^®^ software was employed for printing both plain and mesh shape films (parallel lines, Appendix A). The RS Pro printer parameters were: 100% infill, two shells, and extruder temperature of 180 to 200 °C. Slow printing speed was used, as the printer had only three printing speed settings: fast, standard and slow. The print bed temperature was 30 °C for the RS Pro printer. Printing durations were 90 and 120 s for mesh and plain films, respectively. The printer heads contained 0.4 mm diameter extruder nozzles.

### 4.5. Scanning Electron Microscopy

FEI inspect^®^ scanning electron microscope (SEM) was used to study the surface morphology of Chi-MRs, cellulose microfibres, and 3D printed films at 20 kV accelerating voltage [40]. The dry samples were mounted on aluminium stubs and gold coated using an Emitech K550 (Ashford, UK) sputter coater.

### 4.6. Differential Scanning Calorimetry

Differential Scanning Calorimetry (DSC) analysis (DSC 7; Perkin Elmer^®^, Waltham, USA) was used to analyse the thermal characteristics of powders, filaments, and films with a nitrogen flow rate of 20 mL/min and a heating rate of 20 °C/min. The samples were heated to 220 °C. The indium standard was used to calibrate the system. The least and maximum values of the endothermic and exothermic peaks, respectively, were used to determine the melting (T_m_) and crystallisation temperatures (T_c_) [41].

### 4.7. Fourier Transform Spectroscopy

A Spectrum 100 FTIR spectrometer (PerkinElmer^®^, Shelton, CT, USA) was used to obtain the materials’ FTIR spectra. Under ambient circumstances, the samples were analysed in the 4000–650 cm^−1^ range. The spectra for the formulation powder, filaments, and films were analysed using the Spectrum Express programme [40].

### 4.8. Disintegration Tests

Each 3DP film’s disintegration time was measured in distilled water at 37 ± 0.5 °C using a Copley Scientific disintegration tester DTG 1000 (Copley Scientific, Nottingham, UK). The time was recorded for complete disintegration of each film and passing through the wire mesh. Each formulation was tested in triplicate [4].

### 4.9. In Vitro Dissolution Studies

Using USP dissolution equipment (Varian^®^ VK7010, New Jersey, USA), the drug release characteristics of the films were evaluated. All dissolution experiments were performed in accordance with the British Pharmacopoeia, using the dissolving medium (900 mL potassium phosphate buffer, pH 5.8) at a speed of 50 rpm and at a temperature of 37 °C. The samples were collected with the auto-sampler between 0 and 30 min, at 5 min intervals (i.e., at 0, 5, 10, 15, 20, 25, and 30 min) and analysed with a Cary 50 UV-Vis spectrophotometer at a wavelength of 243 nm. For each formulation, the drug release profile (percent drug dissolved vs. time) was plotted [42]. Each formulation was tested in triplicate.

### 4.10. Evaluating Mechanical Properties of Filaments and Films

Three different texture analysis methods were utilized to measure the mechanical properties of filaments by using TA.XT (Stable Micro Systems Ltd., Godalming, UK). These tests were three-point bend (3PB), stiffness, and resistance [34]. 

Flexibility of filaments were evaluated using a TA-XT-Plus^®^ analyser (Stable Micro Systems, Godalming, UK) and a 3-point bend probe set (Texture Technologies) (Appendix A). This reflects the feed-ability of the filament into the 3D printer. In other words, if filaments are brittle then these will not be able to push the molten polymer through the narrow hole of the nozzle due to breaking in the printer head [43]. At first, filament samples were prepared in 6 cm long pieces. The gap between the plates was 25 mm on the sample holder of the 3-point bend tester. The blades moved at a speed of 10 mm/s until they reached 1 cm under the sample container. Exponent software version 6.1.6.0 (Stable Micro Systems, Godalming, UK) was utilised for data collection and analysis, and samples of each formulation were evaluated [44].

The stifness test indicates the mechanical stability of the filament in the gear mechanism, i.e., not to be squashed by the gears. For the stiffness test, filament samples were prepared with 5 cm length. The samples were positioned on the sample holder’s flat surface (Appendix A). The blade entered the material with a 35% change in shape/deformation (0.6 mm), and data on breaking stress/force were acquired. Each formulation was evaluated in trplicate, and three measurements were conducted for each replica. The data were assessed using the Exponent programme version 6.1.6.0 (Stable Micro Systems, Godalming, UK) [43].

Resistance (compression) tests were carried out with an in-house made rig (Appendix A) and a 5 kg Load cell to imitate the feeding process of a filament through the printer head. Filaments were squeezed axially at 3.15 mm/s, which corresponds to the roller movement speed of a typical FDM 3D printer. To allow bending and avoid fracture with the clamps, 5 cm long filament sections were kept standing beween two syringe filter holes. The compression distance was set to 10 mm with a 0.05 N trigger force, and data were captured during both compression and release. Three filaments were tested from each formulation replica, and each single filament was tested three times (total 9 measurements for each replica) [45].

Tensile behaviour of films was evaluated with the texture analyser (TA-XT-Plus, Stable Micro Systems, Godalming, UK). The films (*n* = 3) were attached between the instrument’s tensile grips and stretched at a speed of 2 mm/s till breaking point, using a trigger force of 0.049N. The tensile properties (film elongations and tensile strength) were calculated from the force-time plots [46].

### 4.11. Content Uniformity

An Agilent 1200 series high performance liquid chromatography (HPLC) (Stockport, Cheshire, UK) was used to analyse the drug content of the films. Mobile phase was a mixture of methanol and water (3:1). Flow rate was set at 1.5 mL/min; detection spectrophotometer was set at 243 nm, retention time was 3 min and sample volume was 10 μL. The stationary phase comprised of a C-18 column (ZORBAX^®^ Eclipse XDB-C18, 4.6 × 150 mm, 5 µm, 400 bar pressure limit manufactured by Agilent^®^, Santa Clara, CA, USA). A calibration curve was prepared for paracetamol with a linear relationship between 0.1 and 1 mg/mL (R^2^ = 0.999). Ten films were analysed from each formulation [4].

### 4.12. Cell Toxicity Evaluations

The human cervical adenocarcinoma cell line HeLa (originally obtained from the European Collection of Authenticated Cell Cultures (ECACC), Salisbury, UK. was cultured in Dulbecco’s Modified Eagle Medium (DMEM, with 4.5 g/L D-glucose) supplemented with 10% foetal calf serum (FCS), 1% L-Glutamine (2 mM), and 1% antibiotic-antimycotic solution. They were grown as adherent monolayer cultures in T75 cm^2^ tissue culture flasks and maintained at 37  °C in a humidified atmosphere of 5% CO_2_ and 95% air. Cells were detached from flasks through trypsinisation with recombinant trypsin (TrypLE). Cell density was determined through haemocytometer-assisted counting under the microscope [47].

To test the cytotoxicity of oral films (F2, F3, F5, F8, F9, F10, F11, F19, F20, F21 and F22), HeLa cells were seeded into 96-well plates at 7.5  ×  10^40^ cells/mL (100 μL/well) and allowed to adhere overnight (24 h) in an incubator at 37 °C with 5% CO_2_. The growth medium in each well was then removed, and the cultures were treated with 100 μL of each sample solution (1 mg/mL), which was made by dissolving 10 mg of individual film in 10 mL of growth medium. After 48 h of incubation, cytotoxicity was assessed using the MTT assay. 10 μL of MTT (5 mg/mL in PBS) was added to each well and the plate was incubated. After 3 h of incubation, the content of each well was aspirated and 100 μL of DMSO was added, and the plate was shaken on an orbital shaker for 3–5 min. The absorbance of the wells was then read at 570nm using a spectrophotometric plate (Spark10M^®^, Tecan, Switzerland). Three duplicates of each treatment were used in each experiment, which was performed three independent times. The viability of the negative control was taken as 100%, and the viability of each treatment was normalised to it [48].

## 5. Conclusions

This work demonstrated that micro-ribbons/microfibres of chitosan or cellulose could change the mechanical properties of filaments produced by hot melt extrusion. Both chitosan micro-ribbons and cellulose microfibres made the filaments unsuitable for printing at high concentrations. Using cellulose microfibres helped produce films with better appearance and chitosan micro-ribbons were able to act as a disintegrant at a low concentration (1% *w*/*w*) compared to a conventional disintegrant for PVA films. This suggests that chitosan micro-ribbons form a network of hydrophilic channels within the film, which helps rapid disintegration of the film in aqueous media. However, this trend was not followed for PVP films. Films containing microfibres/micro-ribbons did not show cell toxicity. The stiffness of printable filaments was comparable to that of unprintable filaments, suggesting that other factors prevented printing with these filaments. FTIR data suggested chemical changes in PVP films when chitosan micro-ribbons were used. Films demonstrated different release profiles when chitosan micro-ribbons or cellulose microfibres were added into the formulation.

## Figures and Tables

**Figure 1 pharmaceuticals-16-00079-f001:**
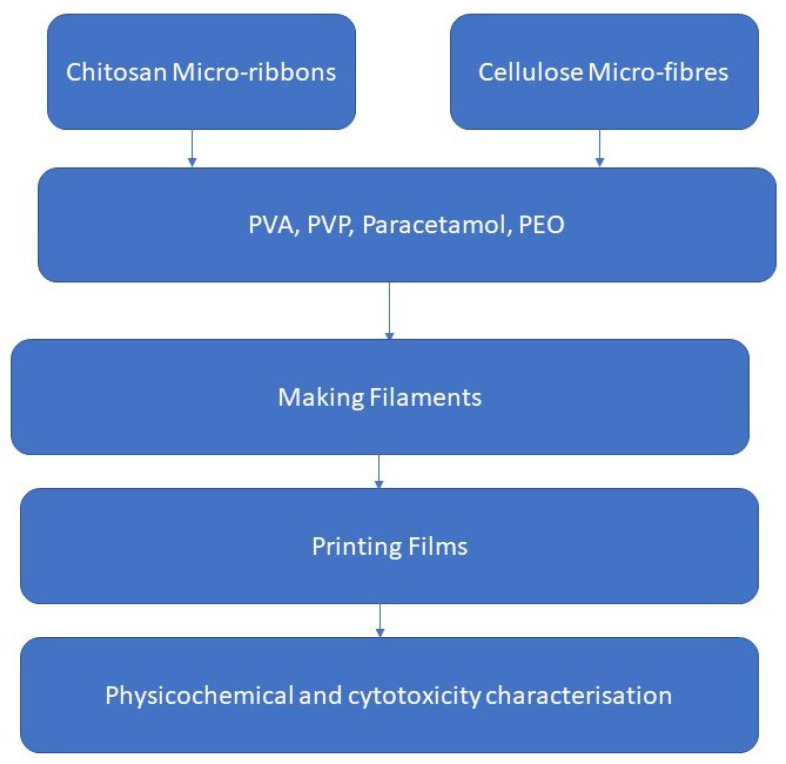
The research plan of this study.

**Figure 2 pharmaceuticals-16-00079-f002:**
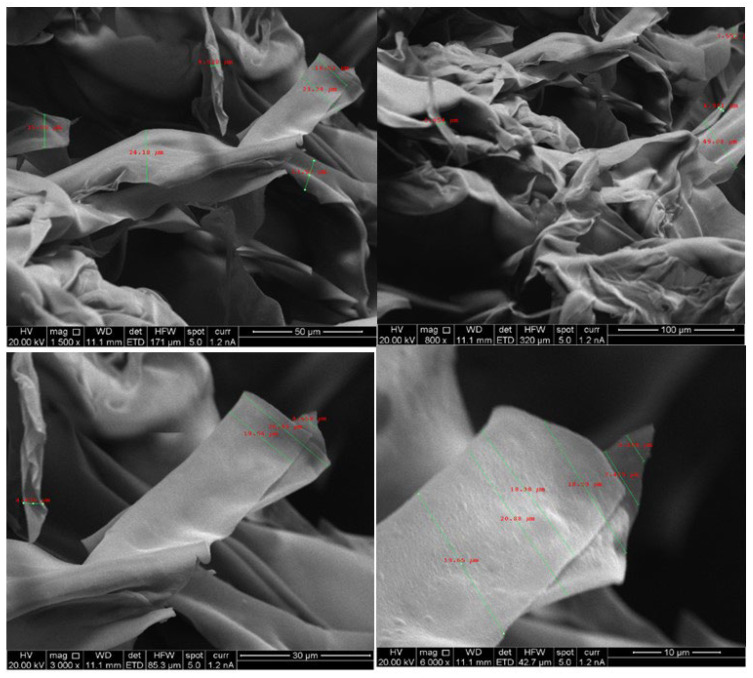
SEM micrograph of chitosan micro-ribbons.

**Figure 3 pharmaceuticals-16-00079-f003:**
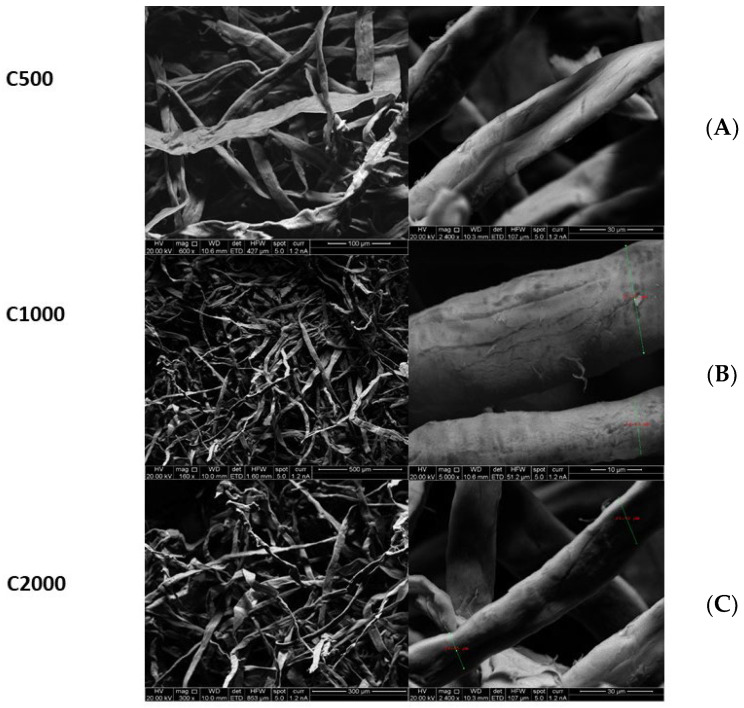
SEM micrograph of cellulose microfibres (Celova^®^) with three different surface areas (**A**): C500 (mm^2^/g), (**B**): C1000 (mm^2^/g) and (**C**): C2000 (mm^2^/g)).

**Figure 4 pharmaceuticals-16-00079-f004:**
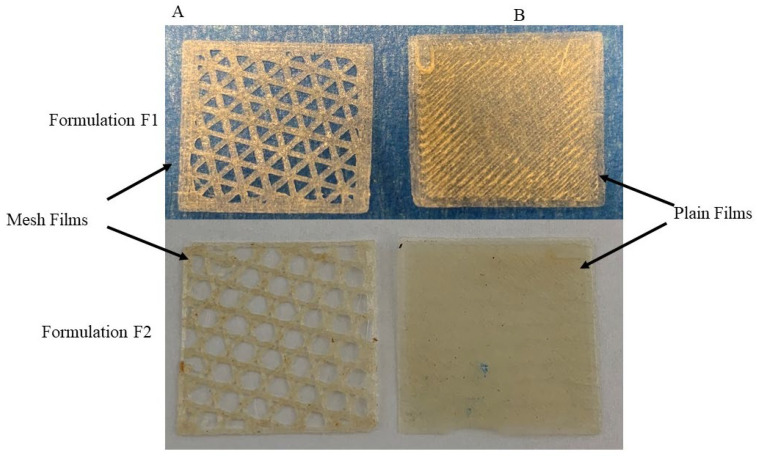
Photographs of typical mesh and plain films by 3D printers (**A**) Prusa (formulation F3), (**B**) Rs Pro (formulation F23).

**Figure 5 pharmaceuticals-16-00079-f005:**
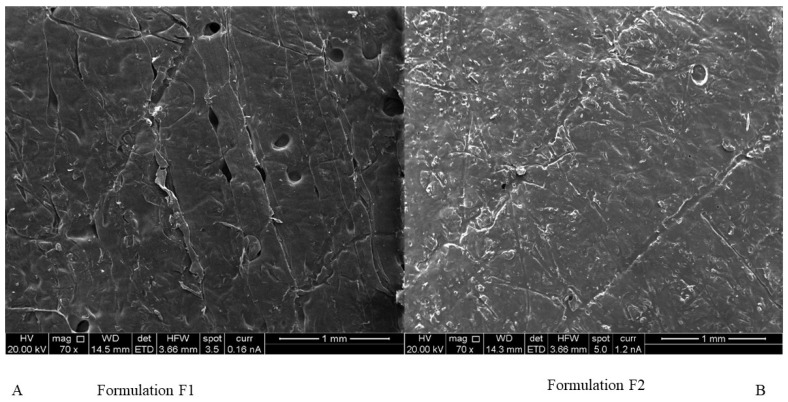
SEM of 3D printed film (**A**) formulation F1 containing no micro-ribbons, (**B**) formulation F2 containing chitosan micro-ribbons at 1% *w*/*w*.

**Figure 6 pharmaceuticals-16-00079-f006:**
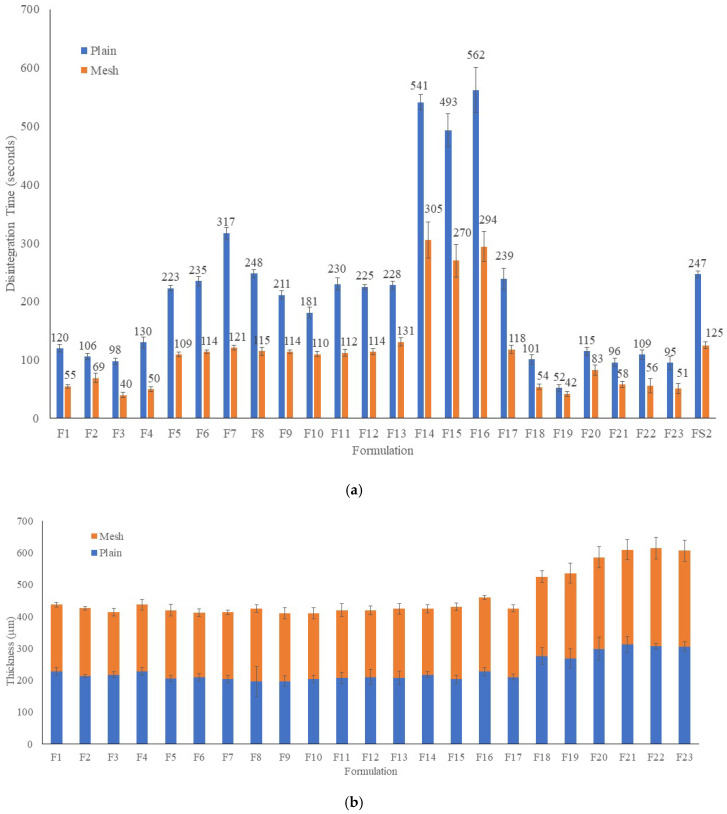
(**a**) Disintegration time of films, (**b**) Thickness of films, (**c**) Weights of films. Data presented as mean ± SD. Error bars indicate SD (*n* = 3).

**Figure 7 pharmaceuticals-16-00079-f007:**
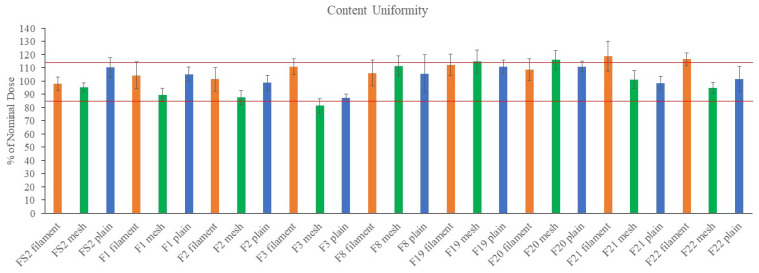
Content uniformity of filaments, mesh films and plain films as percent of nominal dose. The red lines present 85–115% content variation. Data presented as mean ± SD. Error bars indicate SD (*n* = 3).

**Figure 8 pharmaceuticals-16-00079-f008:**
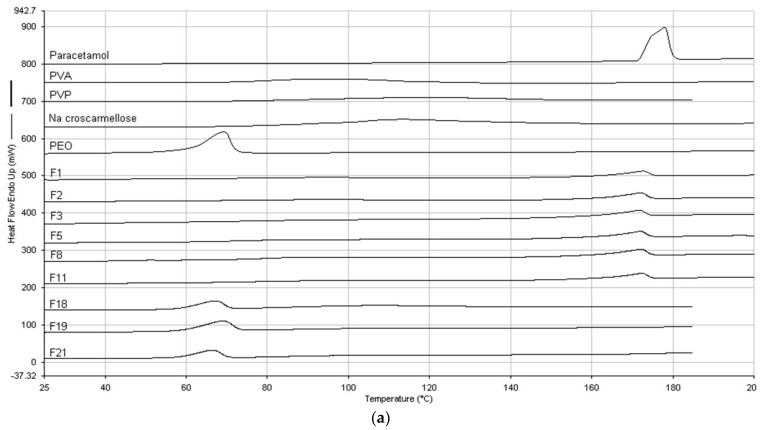
(**a**) DSC thermograms for powder mixtures. (**b**) DSC thermograms of filaments. (**c**) DSC thermograms of films.

**Figure 9 pharmaceuticals-16-00079-f009:**
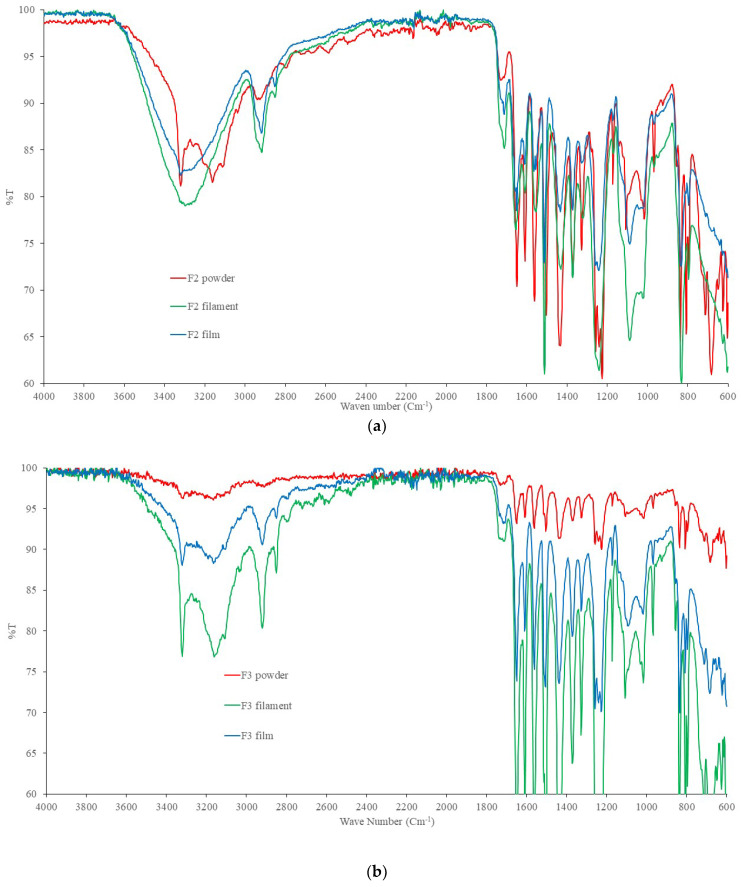
FTIR spectra of powder, filament, and films for formulations (**a**) F2, (**b**) F3, (**c**) F19, (**d**) F20, (**e**) F21.

**Figure 10 pharmaceuticals-16-00079-f010:**
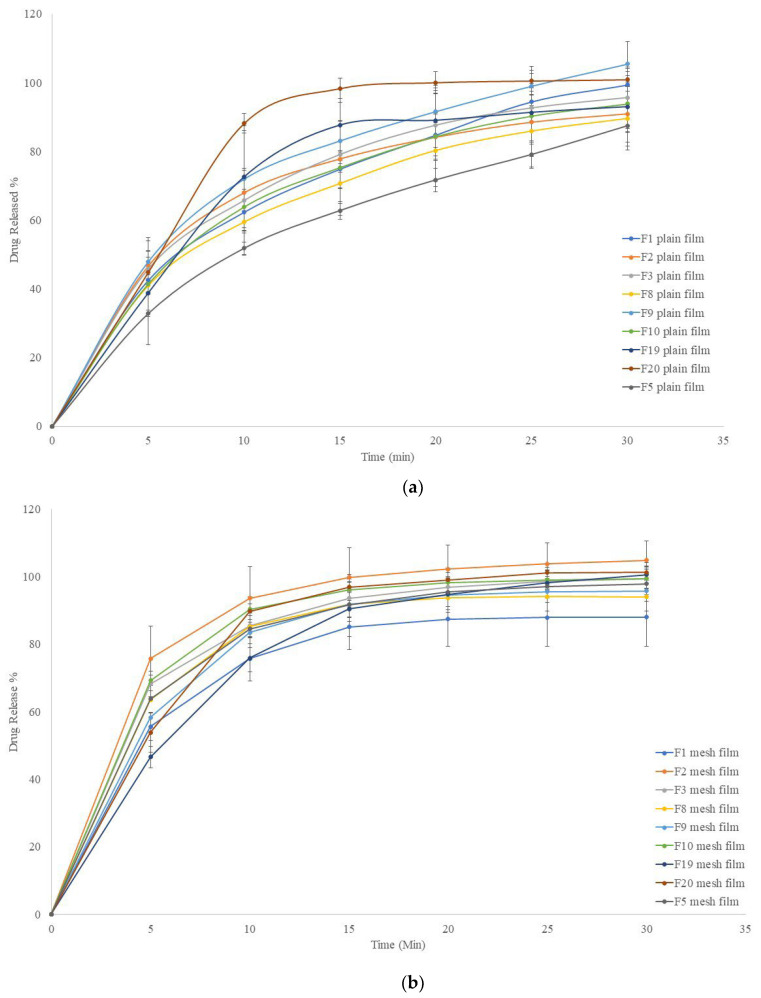
In vitro drug release profiles of (**a**) Plain films, (**b**) Mesh films. Data presented as mean ± SD. Error bars indicate SD (*n* = 3).

**Figure 11 pharmaceuticals-16-00079-f011:**
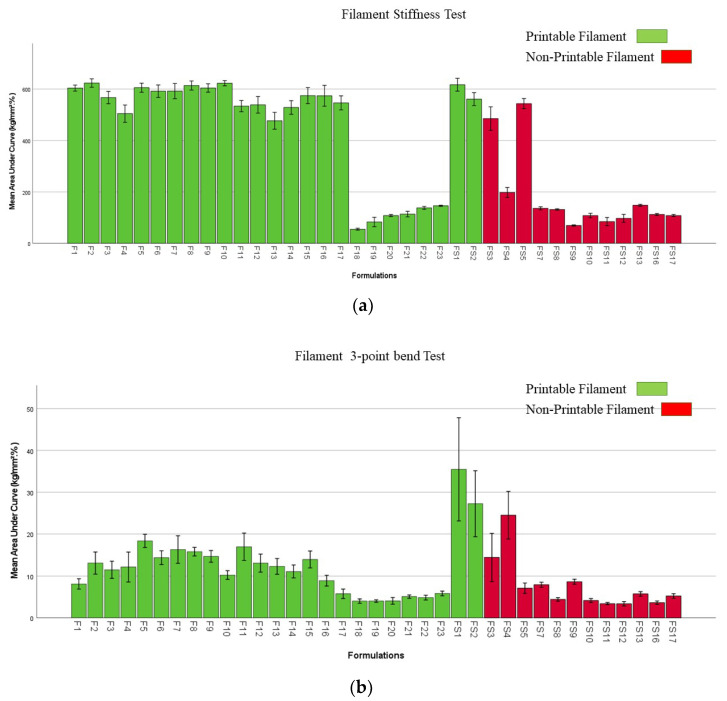
Mechanical properties of filaments: (**a**) Stiffness test (**b**) Flexibility (Three-point bend test), (**c**) Resistance. Data presented as mean ± SD. Error bars indicate SD (*n* = 3).

**Figure 12 pharmaceuticals-16-00079-f012:**
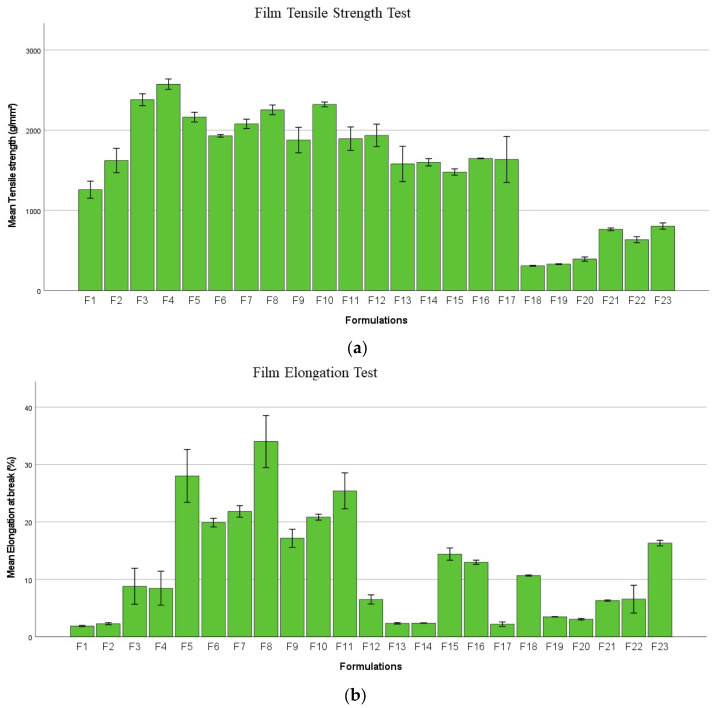
Mechanical properties of films (**a**) Tensile Strength, (**b**) Elongation at break (%). Data presented as mean ± SD. Error bars indicate SD (*n* = 3).

**Figure 13 pharmaceuticals-16-00079-f013:**
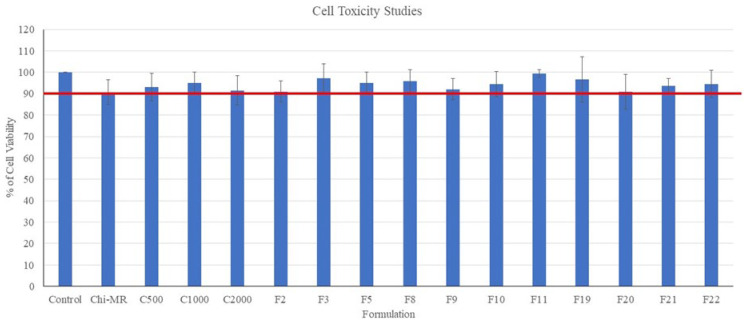
Non-cytotoxicity of films as assessed by the MTT assay after HeLa cells were treated with the films for 48 h. The red lines present 90% cell viability. Data presented as mean ± SD. Error bars indicate SD (*n* = 3).

## Data Availability

Data is contained within the article and Appendix A.

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
