# Peer review of "The Use of Micro-Ribbons and Micro-Fibres in the Formulation of 3D Printed Fast Dissolving Oral Films"

_pharmaceuticals, 2023, doi:10.3390/ph16010079_

Round 1

Reviewer 1 Report (Previous Reviewer 3)

The authors present an article for optimisation of novel fast-dissolving oral films.

The work is very interesting which can be useful for the readers of this journal. this work can be accepted after minor modifications.

- the introduction needs to be improved.

- the quality of the figures must be improved

- The conclusion should briefly cover everything you have discussed in the article.

-The histograms are very cluttered, they must be improved.

- Statistical analysis can improve the quality of your work, especially in the formulation part, for example the use of design of experiment for mixtures to gain in quality and manufacturing cost.

Author Response

We would like to thank reviewer for the comments. Please find below our replies

the introduction needs to be improved.

Thanks for the comment. We improved introduction and removed unnecessary text. We could not use track change for all modifications, as some references were removed, and it was necessary to update the refence list. Otherwise track changes were used.

- the quality of the figures must be improved

Thanks for the comment. We need to draw the attention of the reviewer to the point that the quality of images significantly deceases when PDF is created during the submission process. These will be rectified in the final submission.

- The conclusion should briefly cover everything you have discussed in the article.

Thanks for the comment. The Conclusion section was modified and included topics discussed in the Discussion.

-The histograms are very cluttered, they must be improved.

Thanks for comments. We changed Figure 6B and 6C to stacked bar charts. While, this could not be done for other histograms, as comparing between data would be difficult. Also originally tables included data in Figures 6 and 7. However, the comparison between data was difficult. Histograms allowed rapid comparing data.

- Statistical analysis can improve the quality of your work, especially in the formulation part, for example the use of design of experiment for mixtures to gain in quality and manufacturing cost.

Thanks for your comment. We see the reviewer’s point, and we would like to draw the attention of the reviewer to the point that we applied Taguchi method in our preliminary studies. We observed that not all filaments were printable. Therefore, this statistical approach became unusable.  Then we realised that we need to determine fundamental information about the incorporation of microfibers into the formulation of fast dissolving oral films. The availability of this data would allow applying full-scale design of experiment approach. The design of experiment perhaps can be applied by varying parameters such as infill density, layer height, number of shells, and printing speed. In other words, we should be able to print an object at the first instance, and then by applying the design of experiment improve quality of the printed object. While we noticed that printing became impossible for certain filaments, even with acceptable mechanical properties. Therefore, applying the design of experiment would be the next paper (study) to improve the quality of oral films containing microfibres.

Reviewer 2 Report (New Reviewer)

The manuscript by Marwan Algellay et al., entitled: The Use of Micro-ribbons and Micro-fibres in the Formulation of 3D Printed Fast Dissolving Oral Films, presents an interesting study on the influence of chitosan micro-ribbons or cellulose microfibers on the paracetamol 3D printed fast dissolving oral films.

The conducted research is interesting and is a "must" in the actual stage of the development of 3D printing drugs. It also shows a lot of negative results for some issues which could be considered in such studies. These can offer some light on the domain and can be taken into account in further studies.

I would like to ask the authors to explain the followings:

-        The phrase: The use of high molecular weight polymers may increase the disintegration time, 93 which is not a desired property for FDFs (lines 93-94)

-        Why they didn’t consider the use of an adhesive on the printing platform

-        The choice of the printing temperature, as it can be seen that the 3D printed products are brownish instead of white colored

-        It is understandable that table 1 is presenting the amounts of ingredients as percentages, but still, it should be mentioned

-        the authors declare that SEM analysis was performed also on the 3D printed films (line 163), but no results are presented or discussed

-        lines 270-275 seem to be contradictory and hard to understand. Firstly, it is mentioned that adding only 1% of Chi-MRs to formulations containing PVA allowed printing films, still, SEM micrograph for F1 (without Chi-MRs) proves the contrary.

-        The disappearance of paracetamol peak in the solid dispersions should be discussed more and should be corroborated with the IR findings

-        Finally, The purpose of the films’ use is not clear. Are they meant to be swollen? If so, is the shape helpful? Or are they meant to disintegrate in the oral cavity, a case in which they should have an adequate taste and the disintegration time should be lower and determined in a simulated saliva medium?

Author Response

We would like to thank reviewer for the comments. Please find below our replies

-        The phrase: The use of high molecular weight polymers may increase the disintegration time, 93 which is not a desired property for FDFs (lines 93-94)

Thanks for the comment. We observed longer disintegration time for FDFs in our previous work when we switched from PEO100K to PEO200K. The disintegration time was 103 ± 9 s when PEO100K was used (formulation A in our previous work) and the disintegration time increased to 120 ± 5 s when PEO200K was employed (formulation B in our previous study). Our previous works is cited as reference 4 in the revised manuscript, but it is reference 26 in the original manuscript.

-        Why they didn’t consider the use of an adhesive on the printing platform

Thanks for the comment. We utilised adhesive as sticky masking blue tape (3M™). However, we did not examine other adhesives. This was because the separation of the film from the adhesive becomes an issue, and films are so thin that the structure may get damaged during separation from the adhesive. Our experience has showed so far that sticky masking blue tape (3M™) seems suitable for 3D printing of FDFs.

-        The choice of the printing temperature, as it can be seen that the 3D printed products are brownish instead of white colored

Thanks for the comment. The reviewer is right. We noticed brown films when croscarmellose was utilised as disintegrant. High printing temperatures had to be used, to allow uniform flow of molten polymer from the printer nozzle. When the printing temperature was reduced, the printer failed to print FDFs.

-        It is understandable that table 1 is presenting the amounts of ingredients as percentages, but still, it should be mentioned

      Thanks for the comment. The reviewer is right. The caption of Table is changed to “Table 1. The weight percentage compositions of various ingredients in formulations of FDFs. 3D printing of FDFs was achieved with the filaments of these formulations.”

-        the authors declare that SEM analysis was performed also on the 3D printed films (line 163), but no results are presented or discussed

Thanks for the comments. We would draw the attention of the reviewer to Figures 5A and 5B. These are explained on page 13 paragraph 2 under “3.2.            Formation of Filaments and Films” subheading.

-        lines 270-275 seem to be contradictory and hard to understand. Firstly, it is mentioned that adding only 1% of Chi-MRs to formulations containing PVA allowed printing films, still, SEM micrograph for F1 (without Chi-MRs) proves the contrary.

      Thanks for the comment. The reviewer is right. This line now is changed to “Comparing Tables 1 and S1 shows that formulations F2, F3 and F4 which contained Chi-MRs in the range of 1-2% w/w were printable. Formulations F3 and F4 did not contain croscarmellose.”

-        The disappearance of paracetamol peak in the solid dispersions should be discussed more and should be corroborated with the IR findings

      Thanks for the comment. We understand that the reviewer refers to disappearance of paracetamol in DSC thermograms. We would like to draw the attention of the reviewer to the point that when the peaks of paracetamol disappeared in the thermograms of films and filaments this suggest dispersion of paracetamol in the matrix of filaments and films in molecular levels, as well as possible degradation of paracetamol molecules. Therefore, the reviewer is right and the text on page 16 paragraph 1 was changed to “This implies that throughout the hot-melt extrusion and printing processes, paracetamol diffused in the molten polymer matrix to generate a homogenous solid dispersion, or/and there were changes in paracetamol molecules.”

-        Finally, The purpose of the films’ use is not clear. Are they meant to be swollen? If so, is the shape helpful? Or are they meant to disintegrate in the oral cavity, a case in which they should have an adequate taste and the disintegration time should be lower and determined in a simulated saliva medium?

Thanks for the comment. We would like to draw the attention of the reviewer to the point that the films are designed for fast disintegration in the mouth. This is explained in the first line of the Introduction section. The reviewer is right, and the disintegration time should be low. In fact, this is the aim of this paper. Typically, 3D printed fast dissolving oral films do not disintegrate fast and the disintegration time is much longer than counter parts prepared by the solvent casting method. Our investigations managed to achieve disintegration times around 50 seconds when films were printed as mesh. This is much shorter than other 3D printed films and even films prepared by the hot melt extrusion.

Round 2

Reviewer 2 Report (New Reviewer)

Thank you for replying to all my remarks.

This manuscript is a resubmission of an earlier submission. The following is a list of the peer review reports and author responses from that submission.

Round 1

Reviewer 1 Report

This manuscript described microribbons and microfibers for oral delivery formulation. The authors tried to present many data to support their work. Too many important results are missing. The authors must first clarify the fabrication scheme and structure of nanofibers and nanoribbons. And the disintegration of the formulation in saliva should be checked. Above all, there is no toxicity test for use as a drug formulation. Because PVP can cause severe toxicity in some cases, it is very important to characterize the material and conduct related toxicity tests. Additionally, the authors should present the results of drug delivery into animals or cells. Since the topic itself is very interesting, it is judged that if additional data is included, the quality of the thesis will be improved, but it is judged that it is not appropriate to publish in pharmaceuticals as it is.

Reviewer 2 Report

Comments to authors

The manuscript entitled “The Use of Micro-ribbons and Micro-fibres in the Formulation of 3D Printed Fast Dissolving Oral Films” is focused on the development of the fast dissolving films with paracetamole as a model drug and addition of the chitosan microribbons or cellulose microfibers. The whole idea of the research is rather unclear. There are some works which prove the successful preparation of ODF using the FDM method with PVA. Additionally, the manuscript is illegible due to high number of formulations.  Good research does not need so many of them; it is better to perform more in depth analysis of just a few well-designed formulations.

The list of some major issues is listed below:

-        The introduction of the article does not bring any new information or insights into the 3D printing, personalised medicine, formulating ODFs, or any other. It is just a repetition of the information that potential readers already know. It should be thoroughly changed.

-        The authors stated in the introduction section: “The fused deposition modelling (FDM) 3DP technology is the most frequently used technique to make oral films” – this is not true! Hot-melt extrusion and casting methods are definitely more frequently used.

-        The authors stated in line 97 “The main advantage of 3DP for pharmaceutical application is producing multi-layered FDFs, [25] and achieving personalised FDFs [26]” – this is too general statement which makes it false.

-        The purpose of the study and the research plan is unclear. Maybe some scheme representing an idea can shed some light on the authors plan.

-        The description of the methods is insufficient.

o   The authors used a single-screw extruder, how can the single screw counter- or co-rotate?

o   What means that the extruder was measured after each formulation replica?

o   The 3D printing description should be better described as well (printing nozzle size, printing path width, height, etc.).

-        The tables are definitely too big, and the photograps of produced FDFs are too small.

-        Figure 1 does not add anything to the manuscript.

-        The results are poorly described and are almost not discussed.

-        The order of the results presentation makes doubts why the films (which are made from filaments) are described before filaments results?

-        The authors performed content uniformity studies only for one formulation, why? The content uniformity studies should be carried out with ten samples not three.

-        The SEM pictures are too small, so even the scalebar is illegible.

-        The authors concluded that the addition of high concentrations of microribbons and microfibers could increase the viscosity of the molten polymer during printing but they did not perform viscosity studies, so it is not supported by the results.

-        The entire manuscript is illegible due to the high number of formulations. Only the formulations and the results for them, which were used to draw conclusions should be included in the manuscript. All other can be placed in the supplementary materials.

There are also many minor issues; however, the research and manuscript should be changed in general, so I do not point out all the minor issues.

In my opinion, the manuscript cannot be corrected without changing the research plan or adding new analyses which should change the aim of the study; the whole manuscript should be also rewritten to increase readability.

I recommend the rejection of the article.

Reviewer 3 Report

The idea is innovative, the paper is well prepared, and all the needed characterizations are performed.

I recommend to accept the paper after minor revision:

A photo of the used equipment is to be added.

The film designs are to be detailed and the chosen characteristics are to be justified.

In the SEM micrograph, the legends presenting the dimensions are very small and cannot be read.

The mechanical tests are not well described and discussed.

Some figures have very low resolution.

I propose adding a table presenting the Strengths and weaknesses of the various formulations and kind of films.

The paper is to be checked against misprints